# In-Situ Data-Driven Buffeting Response Analysis of a Cable-Stayed Bridge

**DOI:** 10.3390/s19143048

**Published:** 2019-07-10

**Authors:** Sehoon Kim, Hyunjun Jung, Min Joon Kong, Deok Keun Lee, Yun-Kyu An

**Affiliations:** 1Korea Infrastructure Safety Corporation (KISTEC), Jinju-si 52856, Gyeongsangnam-do, Korea; 2Department of Architecture Engineering, Sejong University, Seoul 05006, Korea

**Keywords:** buffeting responses, cable-stayed bridge, full-scale measurements, measured actual buffeting responses, damping ratios

## Abstract

To analytically evaluate buffeting responses, the analysis of wind characteristics such as turbulence intensity, turbulence length, gust, and roughness coefficient must be a priority. The analytical buffeting response is affected by the static aerodynamic force coefficient, flutter coefficient, structural damping ratio, aerodynamic damping ratio, and natural frequencies of the bridge. The cable-stayed bridge of interest in this study has been used for 32 years. In that time, the terrain conditions around the bridge have markedly changed from the conditions when the bridge was built. Further, the wind environments have varied considerably due to climate change. For these reasons, the turbulence intensity, length, spectrum coefficient, and roughness coefficient of the bridge site must be evaluated from full-scale measurements using a structural health monitoring system. Although the bridge is located on a coastal area, the evaluation results indicated that the wind characteristics of bridge site were analogous to those of open terrain. The buffeting response of the bridge was analyzed using the damping ratios, static aerodynamic force coefficients, and natural frequencies obtained from measured data. The analysis was performed for four cases. Two case analyses were performed by applying the variables obtained from measured data, while two other case analyses were performed based on the Korean Society of Civil Engineers (KSCE) Design Guidelines for Steel Cable Supported Bridges. The calculated responses of each analysis case were compared with the buffeting response measured at wind speeds of less than 25 m/s. The responses obtained by numerical analysis using estimated variables based on full-scale measurements agreed well with the measured buffeting responses measured at wind speeds of less than 25 m/s. Moreover, an extreme wind speed of 44 m/s, corresponding to a recurrence interval of 200 years, was derived from the Gumbel distribution. Therefore, the buffeting responses at wind speeds of 45 m/s were also determined by applying the estimated variables. From these results, management criteria based on measurement data for in-service bridge are determined and each level of management is proposed.

## 1. Introduction

Wind-induced vibration has become a critical issue in engineering fields due to the increasing number of long-span bridges with slender, low-frequency structures. Wind-induced vibration on a long-span bridge can be divided into self-excited vibration, vortex-induced vibration, wake-induced vibration, and buffeting vibration. Among these, the buffeting vibration is regarded as a significant excitation due to the low frequencies characterizing both the input and response vibration of the structure. The buffeting response of bridge due to buffeting vibration can be obtained from wind-tunnel experiments, numerical analysis, and measured data. Wind-tunnel tests have been employed to predict the aerodynamic behavior of a given bridge structure, to estimate important aerodynamic parameters, and to investigate aerodynamic stability. For long-span bridges, wind-tunnel tests typically employ two models (section models and full-structure models) [1] and two stages (construction stage [2] and in-service stage [3]), depending on the aerodynamic characteristics of interest. However, wind-tunnel tests are performed using assumptions, and recreating the actual turbulence around the bridge site is difficult. Moreover, due to the high cost of constructing and testing the models, wind tunnel testing is an inefficient method for evaluating the buffeting response of in-service bridge.

On the other hand, numerical analysis approaches are a more cost-effective method for predicting buffeting responses. Over last three decades, computational fluid dynamics (CFD) and computational structural dynamics (CSD) techniques have progressed rapidly with advances in computer performance and algorithms used for solving fluid–structure interaction (FSI) problems [4]. Either two-dimensional (2D) or three-dimensional (3D) CFD analyses could be conducted for efficiently determining aerodynamic parameters [5,6,7,8]. However, to construct high-precision models that reflect the conditions around a given bridge, the analysis of wind effects such as turbulent intensity, turbulent spectrum, roughness coefficient, and gust factor should be performed first. In addition, deriving reliable analysis results is difficult because such results are influenced by variables such as the static aerodynamic force coefficient, flutter factor, structural damping ratio, aerodynamic damping coefficient, natural frequency, and aerodynamic admittance.

Evaluation of the buffeting response using actual measurement has been conducted for the Humber bridge [9], Akashi-Kaikyo bridge [10], Second Servern Bridge [11], Tsing Ma suspension bridge [12], Runyang suspension bridge [13], Sutong Bridge [14,15,16], Stonecutters bridge [17], and Hardanger bridge [18,19]. In these previous studies, the actual measurements were compared with the predicted response at design stage; however, this method has limitations because the actual response to the design wind speed could not be confirmed if the designed wind speed did not occur. Therefore, the most reasonable method for evaluating the buffeting response to the designed wind speed of an in-service bridge utilizes the measured data to extrapolate key parameters.

In this study, the wind speed measured on an in-service steel cable bridge was used to evaluate the damping ratio, aerodynamic force coefficient, and natural frequency of the bridge site by analyzing the acceleration and displacement of the bridge along with the wind environment such as the highest wind speed expected for 200 years, turbulence intensity, turbulent length, and roughness coefficient. Furthermore, buffeting analysis was performed in the frequency domain by applying the measured wind load and damping ratio, static aerodynamic force coefficient, and natural frequency. The results were compared with the actual response of the bridge, and the most reasonable buffeting response of the steel cable bridge for 200-year expected wind speed was proposed. In addition, reasonable management criteria for maintenance of an in-service bridge were proposed using the results of in-situ data-driven buffeting response analysis.

## 2. Analysis of Wind Characteristics at the Bridge Site

### 2.1. Sensor Locations

As shown in Figure 1, the target bridge of this study is a steel-deck cable-stayed bridge with a total extension of 450 m and a width of 11.834 m. For this study, sensors were installed on the bridge, including displacement meters, accelerometers, and anemovanes. The details of installed sensors are presented in Table 1.

### 2.2. Design Wind Speed

To estimate the design wind speed of the bridge, the wind speed was measured over 24 months (November 2010 to October 2012), and the maximum value of average wind speed for 10 min was used. The maximum wind speed for a 200-year return period was calculated by applying Gumbel’s extreme distribution model [20,21]. The Gumbel probability distribution function has the following double-exponential form:(1)$$f\left(x\right)=\frac{1}{a}\exp\left[-\frac{x-b}{a}-\exp\left(-\frac{x-b}{a}\right)\right]$$
where a and b are parameters representing the scale and location, respectively. These parameters can be estimated using the moment method, maximum-likelihood method, or least-squares method. The following probability density function was then used to obtain the 200-year maximum wind speed:(2)$$F\left(x\right)=\exp\left[-\exp\left(-\frac{x-b}{a}\right)\right]$$

The suitability of the measurement wind speed and Gumbel distribution was verified by the probability plot correlation coefficient (PPCC) test [22]. However, because this estimation was based on relatively short-term wind speed data, the measure–correlate–predict (MCP) analysis method was applied to the long-term wind speeds measured from the weather station near the bridge site in order to validate the design wind speed estimated from the Gumbel distribution model. The MCP analysis was used to predict either the short-term or long-term wind speed at the measurement point by correlating the short-term wind speed data with the wind speed data at the reference point as follows:(3)$$V=\left(\frac{\sigma_{V_{s}}}{\sigma_{U_{s}}}\right)U+{\bar{V}}_{s}-\left(\frac{\sigma_{V_{s}}}{\sigma_{U_{s}}}\right){\bar{U}}_{s}$$

Here, $U_{s}$ and $V_{s}$ represent the wind speed observed at the weather station and at the bridge site, respectively, ${\bar{U}}_{s}$ and ${\bar{V}}_{s}$ are the average of the wind speed measured at the weather station and bridge site during the 10-min measurement period, respectively, and $\sigma_{U_{s}}$ and $\sigma_{V_{s}}$ are the standard deviations of the wind speed data measured at weather station and bridge site, respectively.

Figure 2 shows the MCP calibration equation calculated using Equation (3). The MCP calibration on the pylon and the bridge deck was V=0.93U + 0.37 and V=0.76U + 0.40, respectively.

Table 2 shows the design wind speed obtained using the measurement wind speed and Gumbel distribution and that obtained through MCP calibration. The 200-year wind speed of the weather station was applied with the results of the previous study [23]. The 200-year maximum wind speed obtained the Gumbel distribution and MCP analysis was 44 m/s and 46 m/s, respectively. Therefore, the 200-year wind speed of 45 m/s was used as the design wind speed for the buffeting response evaluation.

### 2.3. Turbulence Intensity and Surface Roughness Coefficient

Because the turbulent characteristics of wind vary depending on terrain conditions, estimations based on actual measurements are preferred. Turbulent intensity is the physical quantity of wind characteristics in the natural condition and can be obtained by the standard deviation of the 10-min wind speed measurement divided by the average wind speed:(4)$$I_{u}\;=\;\frac{\sigma_{u}}{U}$$
where U is the average wind speed, and $I_{u}$ is the turbulence intensity in the direction of the airflow. The guidelines for design of cable steel bridges [24] are given as:(5)$$I_{u}\;=\;\frac{1}{{\ln(30/z}_{0})}\cdot {\left(\frac{30}{z}\right)}^{\alpha }$$

Here, $z_{0}$ is the length of surface roughness, $z_{b}$ is the minimum height, z is the elevation of the considered wind speed, and α is the roughness coefficient.

Figure 3 shows the turbulence intensity obtained using the average wind speed and standard deviation measured for 10 min on the pylon and bridge deck. Figure 3a,b are the turbulence intensities calculated using wind speed data in the direction perpendicular to the bridge on the pylon and deck, respectively. Figure 3c presents the turbulence intensity with respect to the wind direction and shows the change in turbulent intensity according to the terrain characteristics near the bridge. Table 3 shows the turbulence intensity of the wind in the longitudinal and perpendicular directions of the bridge. The average turbulence intensity (pylon: 0.176; bridge: 0.173) in the direction perpendicular to the bridge was lower than that (pylon: 0.243, bridge deck: 0.25) along the longitudinal direction of the bridge. In addition, for buffeting analysis, the maximum value of the turbulent intensity in the direction perpendicular to the bridge was extracted by the wind speed and a suitable exponential function.

The roughness coefficient of the surface was calculated by the vertical distribution of average wind speed following an exponential law:(6)$${\bar{U}}_{1}\;=\;{\bar{U}}_{2}{\left(\frac{z_{1}}{z_{2}}\right)}^{\alpha },\;\alpha \;=\;\frac{\log\left({\bar{U}}_{1}/{\bar{U}}_{2}\right)}{\log\left(z_{1}{/z}_{1}\right)}$$

In Equation (6), ${\bar{U}}_{1}$ is the average wind speed of the pylon,$\;{\bar{U}}_{2}$ is the average wind speed of the bridge deck, $z_{1}$ is the elevation of the anemometer at the pylon (91 m), $z_{2}$ is the elevation of anemometer at the bridge deck (29 m), and α is the roughness coefficient of the surface.

In terms of prevailing wind conditions in the direction perpendicular to the bridge, the roughness coefficient was 0.148 (Table 3), which was less than that for ground roughness II in the cable steel bridge design code [24]. The roughness coefficient in the longitudinal direction of the bridge was higher than that for ground roughness IV in the design code. In addition, the roughness length was 0.09 and 0.965 for the longitudinal and perpendicular directions of the bridge, respectively, which are similar to the values of ground roughness II and IV, respectively. Thus, the wind characteristics in the direction perpendicular to the target bridge were similar to ground roughness II.

The average turbulence intensity for the longitudinal direction of the bridge was 0.295 and that for the top of the pylon was 0.243. On the other hand, the turbulence intensities in the direction perpendicular to bridge and that for the top of the pylon were 0.173 and 0.176, which are lower than those in the longitudinal direction.

### 2.4. Turbulence Length and Turbulent Spectrum

Figure 4 shows the turbulence length calculation results using the measured wind speed data of the target bridge. To calculate the spectrum of the design wind speed, the average wind speed of each section was estimated using an exponential function with the least-squares method. The turbulent length $L_{u}$ was calculated by multiplying the time scale $T_{u}$ by the average wind speed $\bar{U}$:
(7)$$T_{u}=\int_{0}^{\infty }\rho_{u}\left(\tau \right)d\tau ,\;L_{u}=\bar{U}\cdot T_{u}$$

Here, $\rho_{u}$ is the autocovariance function, and u is the direction of the airflow. The time scale was calculated by integrating the autocovariance function $\rho_{u}$ and by applying Taylor’s hypothesis of frozen turbulence, which states that the turbulence maintains its shape when passing through the observation point:(8)$$\rho_{u}\left(\tau \right)\;=\;\frac{1}{\sigma^{2}}\underset{T\to \infty }{\lim}\frac{1}{T}\int_{0}^{T}u(t)u(t+\tau )\mathrm{dt}$$

The concept of turbulence spectrum has been proposed by several researchers. Figure 5 illustrates the turbulence spectrum shown at the bridge site compared with the turbulent spectra proposed by von Karman [25], Kaimal [26], and Davenport [27], which are given in Equations (9)–(11), respectively:(9)$$\frac{{n\cdot S}_{u}\left(n\right)}{\sigma_{u}^{2}}=\frac{{4\cdot f}_{L}}{{\left[1+70.8{\cdot f}_{L}^{2}\right]}^{5/6}},f_{L}\;=\;\frac{{\mathrm{nL}}_{u}}{\bar{U}}$$
(10)$$\frac{{n\cdot S}_{u}\left(n\right)}{\sigma_{u}^{2}}\frac{{200\cdot f}_{L}}{6{\left[{1+50\cdot f}_{L}\right]}^{5/3}},{=\;f}_{L}\;=\;\frac{n\cdot z}{\bar{U}}$$
(11)$$\frac{{n\cdot S}_{u}\left(n\right)}{\sigma_{u}^{2}}\;=\frac{{2\cdot f}_{L}^{2}}{3{\left[{1+f}_{L}^{2}\right]}^{4/3}},f_{L}\;=\;\frac{n\cdot 1200}{\bar{U}}$$

In all of the above equations, n is the frequency, $L_{u}$ is the turbulent length, $\sigma_{u}^{2}$ is the dispersion of the wind velocity fluctuation component, z is the elevation, $\bar{U}$ is the average wind speed, and $S_{u}\left(n\right)$ is the power spectrum. Figure 5a shows the reduced turbulence spectrum compared with the turbulence spectra given by previous studies using the 10-min average wind speed of 20.3 m/s. Figure 5b shows the turbulence spectrum measured at the site when the wind speed exceeded 20 m/s. The von Karman spectrum was found to be the best match with the measured data.

## 3. Dynamic Characteristics of the Bridge

### 3.1. Damping Ratio

To estimate the structural damping ratio, the random decrement technique (RDT) [28] can be used to evaluate the damping ratio from ambient vibration or from random vibration caused by wind load. This technique uses the sum of the expected values by free vibration D(t) and forced vibration R(t) as follows:(12)E[x(t)] = E[D(t)] + E[R(t)]

In a stochastic process in which the external force is expected to be zero in the dynamic motion equation of a structure subjected to a random external force, the solution of forced vibration is also zero. Therefore, if the expectation value matched the peak value of the time domain data, the forced vibration response would be close to zero, and only free vibration will remain. In this study, RDT was applied to 262 samples of 10-min vibration data measured at wind speeds of less than 3.0 m/s. The estimated result of the target bridge structural damping ratio is shown in Figure 6a.

Damping that acts on a structure under wind load can be divided into structural damping and aerodynamic damping [11]. The aerodynamic damping ratio was estimated by first applying RDT to the measured time-series data to estimate the overall damping ratio using the wind speed, and then the structural damping ratio estimated for wind speeds below 3 m/s was subtracted from this estimation. The resulting aerodynamic damping ratios were widely distributed by wind speed. A straight line fit was applied, and the results were compared with the solution of the following equation using pseudo-static theories [29]:(13)$$\zeta_{\mathrm{aero,z}}=\frac{\rho B\bar{U}}{8\rho f_{n}m}{\left[\frac{dC_{L}}{d\alpha }\right]}_{\alpha =0}$$
where ρ is the air density (${\mathrm{kg}/m}^{3}$), B is a girder width (m), $\bar{U}$ is an average wind speed (m/s), α is an angle (°), m is a mass per unit length, $f_{n}$ is a natural frequency, $C_{L}$ is a lift coefficient and, ${\mathrm{dC}}_{L}/d\alpha$ is a function of the angle of the lift coefficient. The estimated damping ratios and fitted lines are illustrated in Figure 6.

In Figure 6b, the dashed line shows the wind tunnel test result of a cross-sectional shape similar to the target bridge, and the dash-dotted line is the application of the derivative function of the lift coefficient by CFD analysis [30].

### 3.2. Static Aerodynamic Force Coefficient

In this study, the horizontal and vertical displacements per unit load were analytically calculated. Using these calculated displacements, the drag coefficient $C_{D}$ and lift coefficient $C_{L}$ were estimated for an applied wind load on the bridge as follows:(14)$$C_{D}\;=\;\frac{d_{\mathrm{Hm}}\cdot 1}{\;\left(\frac{1}{2}{\rho U}^{2}H\right)\cdot d_{H0}}$$
(15)$$C_{L}\;=\;\frac{d_{\mathrm{Vm}}\cdot 1}{\;\left(\frac{1}{2}{\rho U}^{2}B\right)\cdot d_{V0}}$$

Here, $d_{\mathrm{Hm}}$ is the measured horizontal displacement, and $d_{H0}$ is an analytical horizontal displacement by unit load. Moreover, $d_{\mathrm{Vm}}$ is the measured vertical displacement, and $d_{V0}$ is an analytical vertical displacement by unit load.

Figure 7 shows the estimated drag coefficient and lift coefficient compared with similar wind tunnel test results [31]. The drag coefficients estimated from Equation (14) was in the range from 0.543 to 0.358 depending on the angle, while the lift coefficient estimated from Equation (15) was in the range from −0.345 to 0.219. In addition, the drag coefficient estimated from the measured displacement response was less than that from the wind tunnel test [31], but the lift coefficient showed a similar maximum and minimum range to the wind tunnel test results.

## 4. Buffeting Response

### 4.1. Buffeting Analysis Input Variable

The buffeting response was evaluated analytically using single-mode frequency-domain analysis [32,33] and time history analysis [34,35,36], and the results were compared with the actual response from the bridge. In order to identify the reasonable analysis case that predicts the actual buffeting response, two cases based on the measured data (Case I and Case II) and two cases based on the design code (Case III and Case IV) were utilized in the analysis conditions, as shown in Table 4. In Case I, the estimated structural damping ratio, drag coefficient, and lift coefficient were obtained from the measured data, and the von Karman spectrum was applied with the wind environment characteristics on the bridge. Case II considered the aerodynamic damping ratio under the same conditions as Case I. In Case III and Case IV, the von Karman spectrum was applied using the structural damping ratio and the wind environment characteristics described by ground roughness I and II, respectively. Finally, Table 5 and Table 6 summarize the static aerodynamic coefficients and turbulent strength, structural damping ratio, and aerodynamic damping ratio used in the analysis.

Of the other factors affecting the buffeting response, the aerodynamic admittance was not considered, and the platter coefficient was applied in the CFD analysis results of the previous study [36]. The aerodynamic admittance could be applied through the Sears function [37], wind tunnel experiments, and actual measurements. However, when the Sears function was applied, the evaluated buffeting response was less than the actual response [35]. Furthermore, aerodynamic admittance was not considered in this study because wind tunnel tests were not conducted, and the wind pressure was not measured at the bridge site.

### 4.2. Buffeting Analysis Results

The wind load, drag coefficient, lift coefficient, and damping ratio were evaluated based on field measurement data. The buffeting analysis was performed as shown in Table 4, Table 5 and Table 6. The results are shown in Table 7 and Figure 8. Moreover, Figure 9 and Figure 10 show the maximum value of the actual displacement for wind speed and the root-mean-square (RMS) value compared with the buffeting analysis results. Figure 9a and Figure 10a demonstrate displacement in the center of the mid-span, Figure 9b and Figure 10b are to illustrate displacement in the quarter of the mid-span. The maximum vertical displacement, as shown in Figure 9 and Figure 10, was 0.4% of the structural damping ratio given in the design code; in the Case IV analysis conditions, this vertical displacement indicated a response close to the upper limit of the measured displacement. In the Case II analysis conditions, the estimated aerodynamic damping ratio was close to the lower limit of the measured displacement. The fitted curve for the measured displacement at the center of the mid-span most closely corresponded to the Case I analysis conditions, and it was greater than the result of the Case II analysis conditions. This result occurred because the measurement displacement included vertical displacement due to vehicle load. The actual pure buffeting response was expected to be similar to the result of the Case II analysis condition. In cable steel bridge design, the Case III analysis conditions are generally used for buffeting analysis in which the structural damping ratio and the turbulent intensity of ground roughness I are based on the design code and the other analysis parameters are calculated from wind tunnel test results. Furthermore, the results of the other analysis conditions at the design wind speed of 45 m/s are shown in Table 8. The response of Case III was 25% smaller than that of Case IV, 47% larger than that of Case II, and 8% larger than that of Case I. Therefore, the Case III analysis was estimated to be 1.9 times the actual buffering response (Case II), so Case III can be safely used for design without actual measurement data such as characteristics of wind environment and damping ratio.

## 5. In-Situ Data-Based Management Criteria

To consider a safety factor in the design, Case III or Case IV should be used. However, applying the buffeting response is a conservative management criterion for maintenance of an in-service bridge. Moreover, in Case I, the actual displacement can be overestimated because the increase in the aerodynamic damping ratio with increasing wind speed is not considered. Therefore, the displacement predicted by Case II and the trend line for measured displacement are appropriate as management criterion for buffering response.

To validate the aerodynamic stability under the most adverse design conditions, buffeting analysis should be performed using variables from the design code and wind tunnel experiments. However, if the buffeting response is used as the criteria for maintaining in-service steel cable bridges, it is reasonable to apply the minimum value predicted by the analysis. Evaluating the buffeting responses for each wind speed in this study, the minimum value for safe bridge maintenance can be determined by Case II. If the 10-min average wind speed is below 25 m/s, according to the cable steel bridge design code, the 50-cm/s^2^ acceleration serviceability criterion can be converted to a 44-mm displacement. Figure 11 shows the range of management criteria by wind speed using a serviceability criterion of 44 mm for wind speeds below 25 m/s.

If the 10-min average wind speed is greater than 25 m/s, additional safety criteria of the bridge are required. The following equations provide two levels of management criteria for buffeting responses by wind speed, namely proposal criteria level I and II:(16)$$d_{b,\mathrm{level}\;I}\;=\;0.133{\;V}_{m}{}^{2}\;-\;2.586{\;V}_{m}\;+\;25.68\;\geq {\;44\;\mathrm{mm},\;V}_{m}\;\geq \;25\;m/s$$
(17)$$d_{b,\mathrm{level}\;\mathrm{II}}\;=\;0.346{\;V}_{m}{}^{2}\;-\;11.930{\;V}_{m}\;+\;148.4\;\geq {\;44\mathrm{mm},\;V}_{m}\;\geq \;25\;m/s$$
where $d_{b}$ is the buffeting response, and $V_{m}$ is the 10-min average wind speed.

Proposal criteria level I applies the maximum displacement of the Case II analysis based on measurement data Proposal criteria level I gives a sufficient buffeting response for the given wind speed, and the buffeting response can be determined by a careful observation on the bridges. On the other hand, proposal criteria level II applies the maximum displacement of the Case I analysis. This level gives the maximum buffeting response for the target bridge through measurement-based analysis. These responses are proposed as the maximum management criteria for vibration and displacement for bridge maintenance. If the responses exceed the management criteria, further examination for the detection of structural damage is required to determine the aerodynamic stability. Figure 12 illustrates the proposed management criteria of bridge based on measurement data.

## 6. Conclusions

This study evaluated the wind loads at a bridge site, such as the turbulence strength, turbulence length, surface luminance coefficient, and wind speed spectrum, based on measured data from the bridge. In addition, the damping ratio, static aerodynamic force coefficient, and natural frequency were estimated using the structural response data, and the buffering response analysis was performed using these variables. The buffeting analysis results were compared with the measured buffeting response and analyzed to determine a reasonable management criterion for the maintenance of in-service cable steel bridges. The conclusions of this study are as follows:(1)The turbulence intensity at the pylon of the target bridge was greater than ground roughness I for general marine bridges, and the turbulence intensity at the bridge deck was similar to ground roughness II. The average value of the surface roughness coefficient in the direction perpendicular to the bridge was 0.168, which closely agreed with ground roughness II. Therefore, the value of ground roughness II could be applied to the roughness coefficient when the wind speed is calibrated for a given elevation.(2)When considering the measured turbulence intensity and turbulent length, the turbulent spectrum was best matched with the von Karman spectrum [25].(3)The static aerodynamic force coefficient estimated based on the measured data was similar to the wind tunnel test results. The actual structural damping ratio was 1.8 times larger than the structural damping ratio given in the design code.(4)The Case III analysis results, which were obtained by applying only the variables of the design code, were compared with the Case I and Case II results, which were obtained by applying analytical variables based on measured data. The response ratios of Case I and Case II to Case III at the design wind speed of 45 m/s were about 93% and 53%, respectively. In other words, the result of Case III was similar to the analysis condition (Case I) using the actual structural damping ratio and aerodynamic characteristics and was 1.9 times larger than the analysis condition (Case II) using the aerodynamic damping ratio.(5)At average wind speeds above 25 m/s for 10 min, criteria related to bridge safety are required, and the management criteria for buffeting responses by wind speed can be given at two levels: the first level is the buffeting response level that normally occurs on the bridge, and the second level is the maximum level of buffeting response that can occur on the bridge. The criteria both require inspections of the bridge to determine necessary variables.


From the above results, this study confirmed that buffeting analysis using the variables proposed in the design code provides an appropriate safety factor. In addition, the demonstrated in-situ data-driven buffeting response analysis can be used as a management criterion for maintenance and monitoring of the structural integrity of in-service long-span bridges.

## Figures and Tables

**Figure 1 sensors-19-03048-f001:**
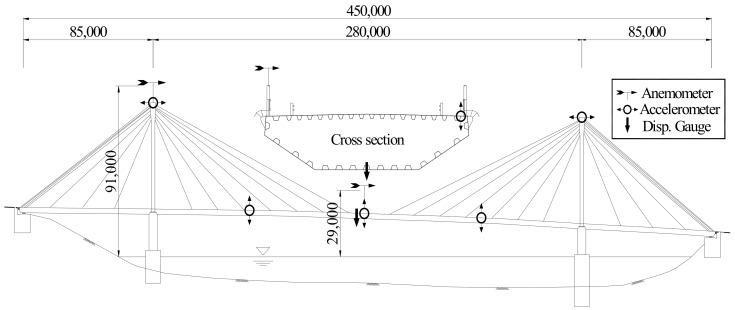
Longitudinal view of the bridge with the sensor positions.

**Figure 2 sensors-19-03048-f002:**
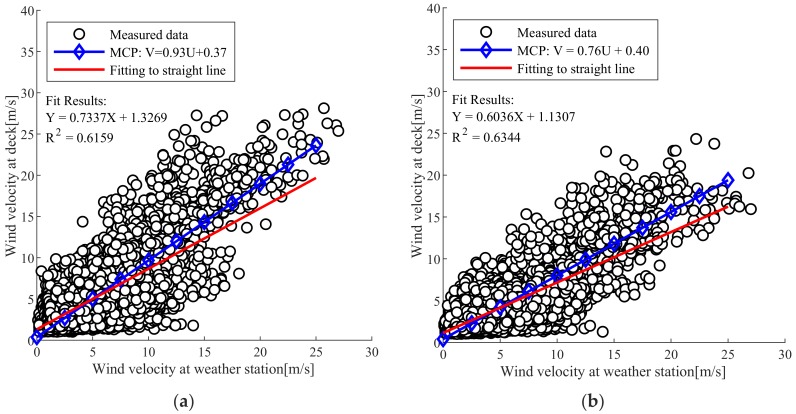
MCP correction of wind velocity; (**a**) At pylon; (**b**) At deck.

**Figure 3 sensors-19-03048-f003:**
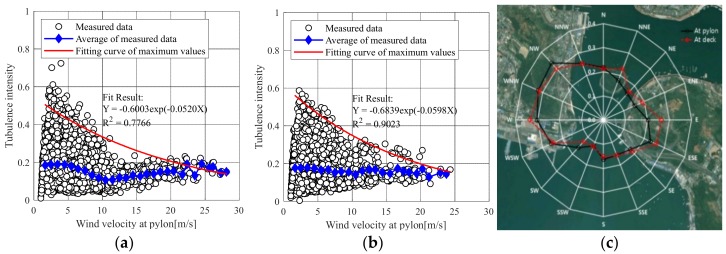
Turbulence intensity: (**a**) At 91 m height; (**b**) At 29 m height; (**c**) Directional turbulence intensity.

**Figure 4 sensors-19-03048-f004:**
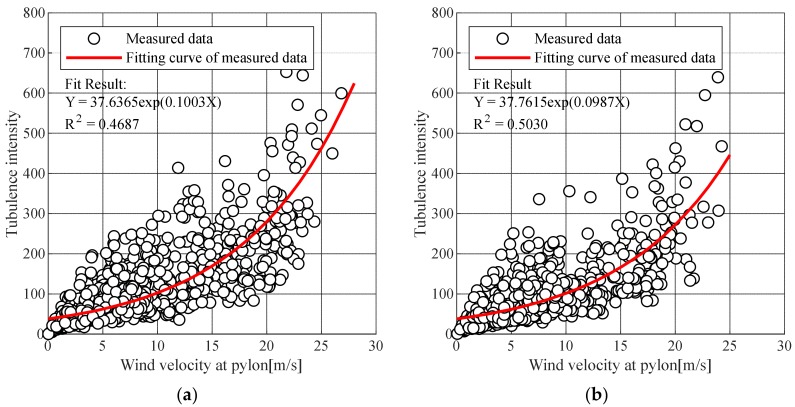
Turbulence length: (**a**) At pylon; (**b**) At Deck.

**Figure 5 sensors-19-03048-f005:**
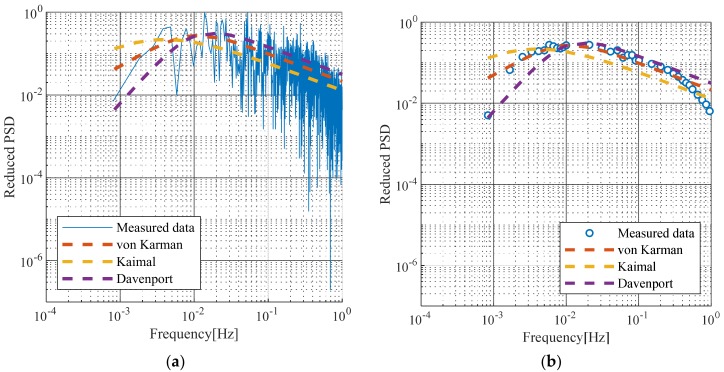
Comparison with turbulence spectra proposed by von Karman [25], Kaimal [26], and Davenport [27]: (**a**) Reduced turbulence spectrum; (**b**) Average reduced turbulence spectrum.

**Figure 6 sensors-19-03048-f006:**
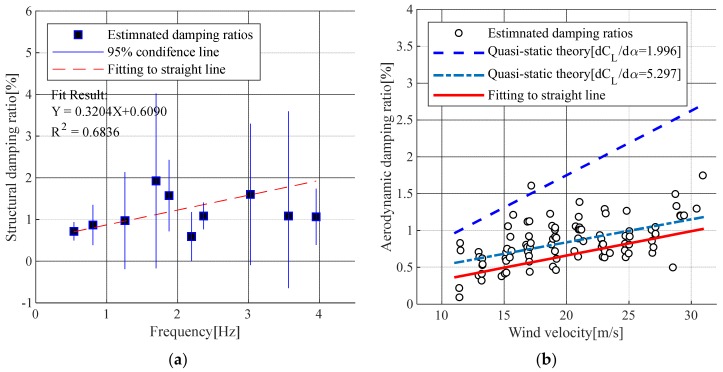
Estimated damping ratios: (**a**) Structural damping ratios of vertical bending modes; (**b**) Aerodynamic damping ratios of the first vertical mode.

**Figure 7 sensors-19-03048-f007:**
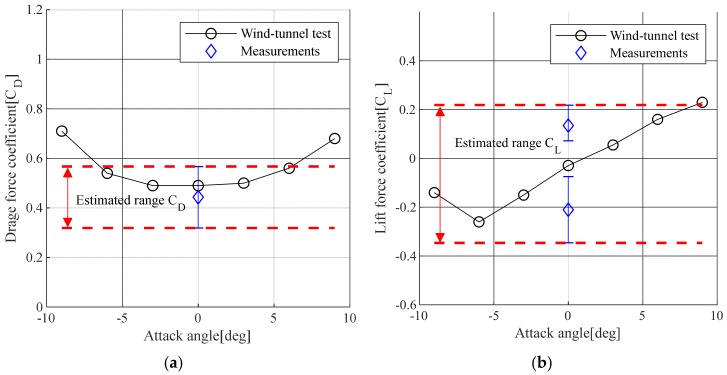
Comparison of estimated values for the drag and lift coefficients: (**a**) Drag force coefficient (C_D_); (**b**) Lift force coefficient (C_L_).

**Figure 8 sensors-19-03048-f008:**
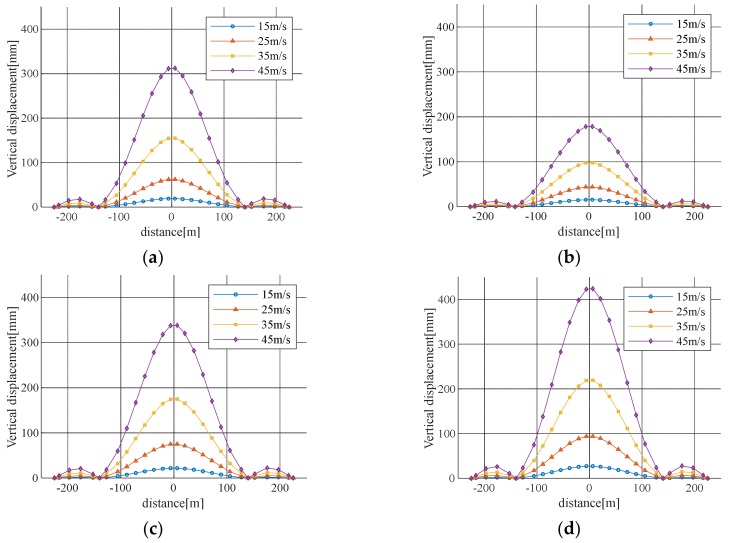
Analytical vertical bending responses at the bridge deck: (**a**) Case I; (**b**) Case II; (**c**) Case III; (**d**) Case IV.

**Figure 9 sensors-19-03048-f009:**
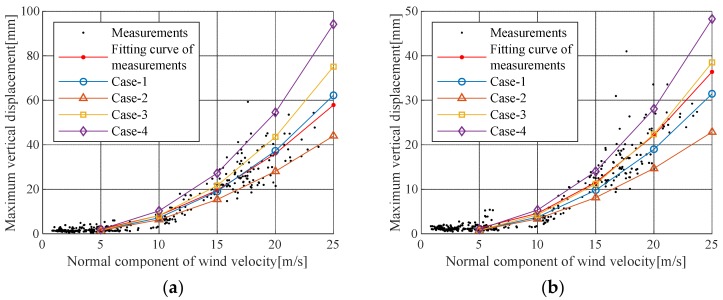
Maximum buffeting responses: (**a**) Responses at the center of the mid-span; (**b**) Responses at the quarter point of the mid-span.

**Figure 10 sensors-19-03048-f010:**
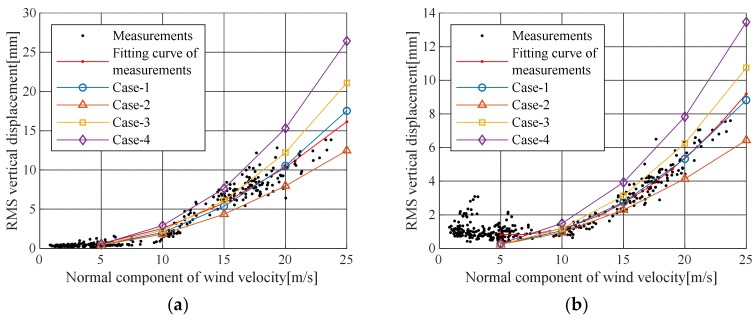
RMS of buffeting responses: (**a**) Responses at the center of the mid-span; (**b**) Responses at the quarter point of the mid-span.

**Figure 11 sensors-19-03048-f011:**
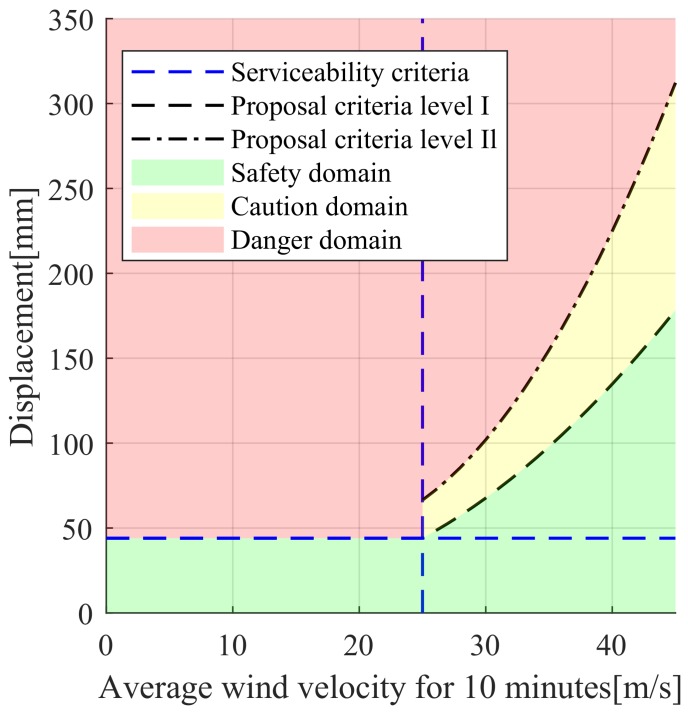
Proposed reference values for buffeting response.

**Figure 12 sensors-19-03048-f012:**
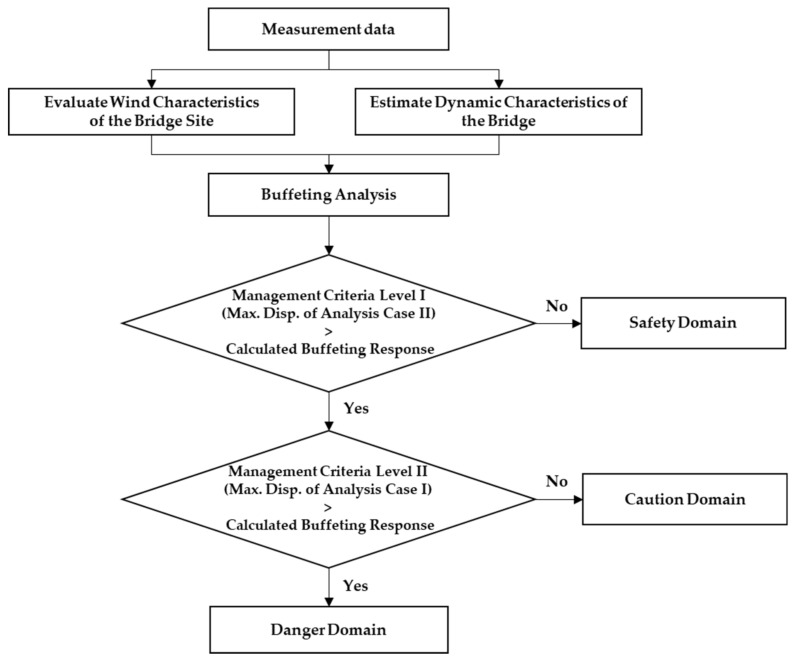
Proposed in-situ data-based management criteria of bridge.

**Table 1 sensors-19-03048-t001:** Specifications of sensors.

Sensor	Quantity	Model	Specifications	Location
Anemometer	2	05103V (R.M. Young)	- Propeller type	Pylon Deck
			- Range: 0~100 m/s, 0° ~ 360°	
			- Threshold Sensitivity: 1.0 m/s	
			- Sampling Frequency: 100 Hz	
Accelerator	2	Kistler 8303A1	- Capacitive Accelerometers	Pylon
			- Range: ±1 g	
			- Resolution: 0.2 mgrms	
			- Sampling Frequency: 100 Hz	
Accelerator	3	Kistler 8303A1	- Capacitive Accelerometers	Girder
			- Range: ±1 g	
			- Resolution: 0.2 mgrms	
			- Sampling Frequency: 100 Hz	
Displacement gauge	1	PSM-LR (Noptel)	- Measuring distance: 600 m	Girder
			- Resolution: 5 mm	
			- Prism: 4P38	
			- Sampling Frequency: 100 Hz	

**Table 2 sensors-19-03048-t002:** Estimated Wind Velocity for a Recurrence Interval of 200-years.

Method	Wind Velocity (m/s)			PPCC	MCP Correction (m/s)	
	M.O.M ^1^	L.S.M ^2^	M.L.E ^3^		Velocity	Reference [23]
Pylon	44	40	44	0.972	46	49
Deck	35	31	34	0.966	38	45

^1^: the method of moment; ^2^: the least-squares method; ^3^: the maximum likelihood estimation.

**Table 3 sensors-19-03048-t003:** Turbulence intensity, roughness coefficient, and length.

Classification	Turbulence Intensity		${\bar{U}}_{1}/{\bar{U}}_{2}$	Roughness Coefficient	Roughness Length
	Pylon	Deck			
Perpendicular	0.176	0.173	1.184	0.148	0.090
Longitudinal	0.243	0.295	1.557	0.387	0.965

**Table 4 sensors-19-03048-t004:** Measured bending frequencies and estimated structural damping ratios.

Mode	Frequency (Hz)	1/4 Point of Mid-Span	1/2 Point of Mid-Span	3/4 Point of Mid-Span	Average
1	0.537	0.746%	0.695%	0.708%	0.716%
2	0.805	0.878%	-	0.856%	0.867%
3	1.245	0.966%	0.965%	0.989%	0.973%
4	1.659	1.922%	2.031%	1.827%	1.927%
5	1.879	1.670%	-	1.477%	1.573%
6	2.196	0.593%	0.590%	0.594%	0.592%
7	2.367	1.093%	1.046%	1.118%	1.086%
8	3.002	1.633%	-	1.571%	1.602%
9	3.563	1.347%	1.368%	1.713%	1.476%
10	3.953	1.117%	1.076%	0.998%	1.064%

**Table 5 sensors-19-03048-t005:** Analysis cases.

Case	Analysis Conditions
Case I	Measured structural damping, drag, lift force coefficients
	von Karman spectrum with measured $I_{u}$ and turbulence length
Case II	Measured aerodynamic damping, drag, lift force coefficients
	von Karman spectrum with measured $I_{u}$ and turbulence length
Case III	Structural damping (0.4%) proposed by design code
	von Karman spectrum with $I_{u}$ of design code (ground roughness I)
Case IV	Structural damping (0.4%) proposed by design code
	von Karman spectrum with $I_{u}$ of design code (ground roughness II)

**Table 6 sensors-19-03048-t006:** Static force coefficients obtained from measured data.

Coefficient	Drag (H)	Lift (B)	Moment
$C_{i}\left(0\right)$	0.444	0.145	−0.043
$\partial C_{i}\left(0\right)/\partial \alpha$	0.395	1.996	−0.344

**Table 7 sensors-19-03048-t007:** Parameters for buffeting analysis.

Coefficient	Case I	Case II	Case III	Case IV
Turbulence intensity	U < 25 m/s: I_u _= 0.173		0.125	0.157
	U ≥ 25 m/s: I_u_ = $0.6839\times e^{-0.05977\times \bar{U}}$			
Structural damping ratio	1st mode: 0.716%		0.400%	0.400%
	2nd mode: 0.867%			
	3rd mode: 0.973%			
	4th mode: 0.3469 × f + 0.5517			
Aerodynamic damping ratio	-	0.0312 × U + 0.2156	-	-
Turbulence length	${47.231e}^{0.087\times \bar{U}}$		200 m	200 m

**Table 8 sensors-19-03048-t008:** Analytical vertical responses at the center of the mid-span (unit: mm).

Wind Speed	Measured		Case I		Case II		Case III		Case IV	
	Max	RMS	Max	RMS	Max	RMS	Max	RMS	Max	RMS
5 m/s	3.9	0.6	1.6	0.4	1.5	0.4	1.6	0.5	2.0	0.6
15 m/s	24.7	5.8	19.1	5.4	15.4	4.4	21.7	6.1	27.2	7.6
25 m/s	66.5	16.1	62.3	17.5	44.0	12.5	75.1	21.1	94.2	26.4
35 m/s	-	-	155.1	43.6	97.9	27.8	174.9	49.0	219.2	61.5
45 m/s	-	-	312.3	87.8	178.4	50.6	338.3	94.9	424.1	118.9
			(0.92)	(0.93)	(0.53)	(0.53)	(1.00)	(1.00)	(1.25)	(1.25)

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
