# Peer review of "In-Situ Data-Driven Buffeting Response Analysis of a Cable-Stayed Bridge"

_sensors, 2019, doi:10.3390/s19143048_

Reviewer 1 Report

Good job!

HOW MANY SENSORS WERE INSTALLED?

 please add a table instead of just mentioning it.

the table should include

1) sensor type 

2) technique used

3) reference 

4)assumption/limitations

the sampling rate for these sensors? which sensors, plz add details.

any overall assumption?

were there any similar studies done in the past? maybe cite some articles that review SHM.

the article is informative, but some times confusing. 

for example, aerodynamic admittance (page 10) was not considered?  how much it effects? there is some error on the page. check it.

Reviewer 2 Report

Structural health monitoring system provides large amount of data to uncover the structural vibration features under wind loads and traffics. This manuscript carried out a comprehensive study on the buffeting responses of a cable-stayed bridge based on the in-situ data recorded by its structural health system, which can provide valuable information for maintenance of in-service bridges in windy areas.

To the knowledge of the reviewer, the structural damping is subjected to the vibration amplitude, i.e., the structural damping increases with the augment of vibration amplitude due to the increase of the friction damping. In this manuscript, the authors just took the identified damping ratios as structural damping when the wind speed is lower than 3m/s. In addition, the excitation forces on the bridge generally include wind and traffic loads. During the low wind speed period, the traffic contributes more important on the vibration than the wind loads. In this manuscript, is it correct to directly compare the measured results with the calculated results? Because of the above-mentioned concerns, the recommended management criteria are suggested to be validated. 

Minor comments:

1. The authors state that the sensors installed on the bridge includes displacement meters, accelerometers, and an anemometer in lines 88 to 89. However, there are two anemometers shown in Fig. 1. Please clarify this.

2. The authors state that the sampling frequency of the sensors was 100 Hz. If sampling frequency of all the sensors including anemometers was 100 Hz. If so, please clarify the type of the anemometers installed on the bridge.

3. As for the data collected by the anemometers, the wind flow may be affected by the structure. How to prevent this barrier effect when collecting the wind data?

4. In line 95, the authors state that the maximum value of average wind speed for 10 minutes was used. It is suggested that the authors use the maximum value of average wind speed for one month and compared with the results for 10 minutes.

5. Some field measurements have indicated that the measured turbulent wind power spectrum in high frequency region is larger than that of the Kaimal, von karman or other experience wind power spectrum. How to explain the opposite phenomenon shown in Fig. 5 in this work.

6. If the buffeting analysis is analyzed in time-domain? More information should be included to clarify the method utilized in the buffeting analysis.

7. Some related references are suggested to have a better introduction part:

Comparative study on buffeting performance of Sutong Bridge based on design and measured spectrum

Investigation of dynamic properties of long-span cable-stayed bridges based on one-year monitoring data under normal operating condition

Long-term monitoring of wind characteristics at Sutong Bridge site

Author Response

Round  2

Reviewer 2 Report

The authors simply take the identified damping ratio, when the wind speed is lower than 3m/s, as the structural damping. To the knowledge of reviewer, the structural damping is a time-changing variable related with the current vibration amplitude which reflects the energy consuming level. In another word, the structural damping is also proportional to the wind speed like the aerodynamic damping. More detailed calculation and descriptions are required to improve the vilidity.
